# Sexuality, Quality of Life, Anxiety, Depression, and Anger in Patients with Anal Fissure. A Case–Control Study

**DOI:** 10.3390/jcm10194401

**Published:** 2021-09-26

**Authors:** Antonio Navarro-Sánchez, Paloma Luri-Prieto, Antonio Compañ-Rosique, Ramón Navarro-Ortiz, María Berenguer-Soler, Vicente F. Gil-Guillén, Ernesto Cortés-Castell, Felipe Navarro-Cremades, Luis Gómez-Pérez, Carla Pérez-Tomás, Antonio Palazón-Bru, Angel L. Montejo, Virtudes Pérez-Jover

**Affiliations:** 1Faculty of Psychology, Miguel Hernández University, 03202 Elche, Spain; navarrosancheztony@gmail.com (A.N.-S.); mariia_269@hotmail.com (M.B.-S.); v.perez@umh.es (V.P.-J.); 2Department of Surgery, San Juan University Hospital, 03550 San Juan, Spain; palomalurip@hotmail.es (P.L.-P.); af.company@umh.es (A.C.-R.); luisgope@gmail.com (L.G.-P.); carlotilla@yahoo.com (C.P.-T.); 3Department of Pathology and Surgery, School of Medicine, Miguel Hernández University, 03550 San Juan, Spain; vte.Gil@gmail.com (V.F.G.-G.); ernesto.cortes@umh.es (E.C.-C.); felipe.navarro@umh.es (F.N.-C.); antonio.pb23@gmail.com (A.P.-B.); 4Orthopedic Surgery and Traumatology Service, Torrevieja University Hospital, 03186 Torrevieja, Spain; ray_hard_@hotmail.com; 5Psychiatry Service, Clinical Hospital of the University of Salamanca, 37007 Salamanca, Spain; 6Institute of Biomedical Research of Salamanca (IBSAL), Paseo San Vicente SN, 37007 Salamanca, Spain; 7Department of Psychiatry, Nursing School, University of Salamanca, Av. Donantes de Sangre SN, 37007 Salamanca, Spain

**Keywords:** anal fissure, sexuality, quality of life, anxiety, depression, anger, questionnaire

## Abstract

Anal fissures (AFs) are lesions located in the lower anal canal. They can be primary (chronic or acute) or secondary to a basic disease. There is high comorbidity of depression and anxiety in patients with chronic AF, with poorer quality of life (QoL) and sexual function. This is a case–control study carried out in the San Juan Hospital (Alicante, Spain). Sixty-seven participants were included in the study, including 35 cases and 32 controls: 36 males and 31 females. This study aims to investigate the association of presenting AFs with sexuality, quality of life, anxiety, depression, and anger. The instruments used were the Spanish validated versions of the validated original selected questionnaires. These instruments were used to assess health-related quality of life, anxiety, anger, depression, and sexual function. Results show higher values in cases than in controls with statistical significance in anxiety state and trait; anxiety and depression; bodily pain, general health, and vitality; and 10 of the 12 anger factors. Higher values in controls than in cases with statistical significance in sexuality and many of the QoL factors were found. Addressing these issues in AF surgical patients would be beneficial for their clinical assessment and intervention.

## 1. Introduction

Anal fissures (AFs) are lesions located in the lower anal canal. They are divided into primary (chronic or acute) and secondary ones; primary AFs develop as local entities; secondary AFs develop in the context of a basic disease [1,2]. AF is a painful longitudinal defect in the anoderm between the anal verge and the pectinate line. AF often causes a reactive hypertonia of the anal sphincter, associated with venous stasis. AFs can heal either spontaneously or after therapy. They can recur as well as turn from acute into chronic ones [1,2,3]. Distribution by age shows a higher incidence in patients from 30 to 50 years old; women and men are both nearly equally affected [1,2,3]. The duration from the first symptoms until the start of treatment such as sphincterotomy lasted more than 10 months on average (Rank from 1 to 96 months) [1]. Chronic AFs [2] can present secondary morphological changes, such as cryptitis, sentinel tags, hypertrophied anal papillae, or anal fistulae [1,2,3]. Secondary AF is associated with other pathologies including ulcerative colitis, Crohn’s disease, and those induced by infectious agents such as syphilis and other *Sexually Transmitted Infections* (STIs) [1,3]. The etiopathogenesis of AFs includes anal sphincter hypertonia, anal sphincter fibrosis, anal abuse, anal sexual practices, infection, ischemia, and traumata. Signs and symptoms include anorectal pain, anal bleeding, anal pruritus, and constipation. AFs’ clinical evolution shows a tendency for recurrence or chronicity [1,2,3].

A chronic AF is defined as a fissure that has been present for longer than 6 weeks. An acute fissure is defined as a fissure that has been present for less than 6 weeks [2]. Arısoy et al. [4] studied patients with chronic AF and age–sex matched healthy controls; both groups were evaluated for psychopathology such as depression, anxiety, and stress through depression, anxiety, quality of life (QoL) and other scales.

There is a high comorbidity of psychopathology in patients with chronic AF [4]. Depression and anxiety severities show a negative impact on the QoL [4]. Stress acts in those with chronic AFs as both a triggering and an exacerbating factor [4]. 

Treatment for chronic AF includes conservative therapy using different drugs [5] and surgical treatment with various surgical techniques [1]. 

Successful treatment of chronic AF leads to improvement in health-related quality of life (HRQoL) measured by a Short-Form (SF) Health Survey, including bodily pain, mental health, vitality, and general health [5].

Modern practice of surgery, beyond surgical techniques, can include global perspectives of the patient as a whole that may often significantly contribute to better patient outcomes [6]. The preoperative optimization of the patient incorporates as a main goal the targeting of social and behavioral factors that can influence recovery [6]. Preoperative optimization also extends to include disease-specific and lifestyle-related modifications in patients with comorbidities [6]. 

Diseases such as AFs and others characterized by genital, anal, or perianal ulcers can be associated with sexually transmitted diseases /infections (STDs/STIs) in symptomatic patients [7,8]. 

Anorectal STIs in sexually active patients are commonly the result of anal receptive intercourse and can be due to contiguous spread from a genital infection [9]. 

Human sexuality is a key function of people; its physical, psychological, emotional, dyadic, and social aspects permeate into many parts of human lives [10]. Sexuality is a major contributor to QoL [10].

Sexual health (SH) includes sexuality-related physical, mental, emotional, and social well-being [11,12]; SH is highly relevant for people’s life fulfillment and is a main component of human QoL, both general and health related [11,12].

Sexuality and health are intimately entwined; sexual functioning may impact dimensions of general health both positively and negatively [13]. Sexual functioning may be either facilitated or disrupted by dimensions of general health [13]. Mental diseases such as depression constitute major risk factors (RFs) for sexual dysfunctions (SDs) impairing sexual life in all domains, including desire, arousal, orgasm, and satisfaction [13]. Somatic symptoms such as chronic pain and physical impairment may also be a RF affecting sexual life and functioning in different domains [13]. Medication used for both mental disorders and somatic symptoms can also negatively affect human sexuality [13]. In clinical practice, it is essential to be able to identify [14] patient-related sexual questions and to assess the influence [14] of related factors to be able to better address it. The relationship between sexuality and mental health is bidirectional [15]. Mental disorders and psychotropic treatments can impact on patients’ sexual functions with implications for clinical management [15,16,17,18,19].

Screening for and managing comorbid depression is strongly recommended [16,17,18,20]. There is a wide range of health benefits related to sexual activity such as some physical and mental health ones, e.g., the potential benefit on pain relief, stress reduction, muscle relaxation, and easier sleep. In a wider sense, sexual activity benefits QoL [21]. Sexual behavior changes through the lifespan depending on the individual’s health, social interaction, and choice of partners [22]. Not only the disease itself but the treatment prescribed may also cause sexual impairment [23]. Biological, psychological, interpersonal, and cultural factors combine in various proportions affecting sexual behavior [24]. Sexual function and dysfunction are regulated by neurobiological foundations of clinical relevance such as hormones and neurotransmitters [25]. 

The quality of life (QOL) of patients with CAFs can be affected by the disease, leading to a marked decline in QoL. QoL can be evaluated by general QoL Questionnaires and by health-related QoL (HRQoL) questionnaires such as SF-36, SF-12, and others [6,7,8]. The QoL of patients diagnosed of chronic AFs can be improved after AF treatment [26]. Chronic medical and psychological conditions and their treatments can be associated with sexual dysfunction and other changes in the sexual life of the patients, including deleterious effects on pleasure, desire, arousal, and orgasm [27]. The prevalence of sexual dysfunction (SD) in patients with a chronic disease is higher than that in a control group. These SDs can entail more emotional and interpersonal serious consequences and can impact on the subject’s QoL [27]. Individuals with medical illness can experience depression, fatigue, pain, stress, and anxiety, which may further contribute to sexual dysfunction [28]. 

Living with a chronic disease affects quality of life, including physical, psychological, and social aspects [29]. These aspects are closely connected, and if one part is affected, it will most certainly influence the others [29]. Chronic physical illness can have a devastating impact for the individual, family, and significant others. Illness over time carries an additional burden upon the individual and their psychosocial world [28]. In our clinical practice, we have detected certain frequent psychological characteristics among patients with anal fissures (AFs). These characteristics include dysfunctional sexuality; lower quality of life (QoL); and a significative presence of anxiety, depression, and anger. Anxiety includes two concepts: state anxiety, such as immediate and current, and anxiety traits such as disposition. Anger expression is including both anger state and anger trait. Our aim was to investigate these characteristics of patients with AFs (cases group) and to compare their results with those of the control group without AFs. Papers covering these aspects are very scarce. This study aims to add to the literature the expected different result profiles between case and controls, of clinical interest.

Our research hypothesis (H1) is that patients diagnosed with AFs (cases group) have a differential profile in certain psychological characteristics with respect to a control group. The null hypothesis (H0), as opposed to our H1, indicates the absence of differences between both groups, cases and controls. For this research, we used a set of specific validated questionnaires.

This case–control study aims to investigate the association of presenting AFs with sexuality, quality of life, and associated psychological factors in a Spanish clinical sample, from a patient-centered perspective, considering them as a whole person.

## 2. Experimental Section

### 2.1. Material and Methods

#### 2.1.1. Study Population

This study involved outpatients diagnosed with idiopathic AFs, acute or chronic, at San Juan de Alicante University Hospital (Alicante, Spain), Alicante-San Juan Health Department 17 of the Regional Ministry of Universal Health and Public Health of the Valencian government [30]. This health department serves a total of 227,767 inhabitants including 115,784 women (53.03%) and 106,983 men (46.97%) (data referred to on 30 December 2019), providing free and universal health coverage [31].

This case–control study [32,33,34] was carried out from January 2016 to February 2017 [3]. Cases were defined as patients diagnosed with acute or chronic idiopathic AFs during the study period who wished to participate in the study and provided written informed consent for this purpose. 

Fissures were considered chronic when symptoms were present for more than 6 weeks or when signs of chronicity were present. Excluded from the study were patients with fissures secondary to other conditions, functional alterations (incontinence or constipation), rectocele or prolapse, or those having undergone previous surgical treatment for anorectal disease or obstetric trauma and having previous clinical diagnosis of a severe psychiatric disorder, specifically, depressive disorder or anxiety disorder [3].

#### 2.1.2. Study Design and Participants

This case–control study [3,33] involved outpatients diagnosed with acute or chronic idiopathic anal fissures [3]. The variables gender (male, female) and age (years) are presented with respect to the total sample, the cases, and the controls in Table 1.

The sample consisted of 67 people, 35 of them (52.2%) were assigned to form the group of cases and the other 32 to form the control group (47.8%). The average age of the participants was 52.2 years (*SD* = 12.1), with 53.7% men and 46.3% women. 

Cases and controls groups description [3]: Patients of the case group were being seen at the hospital surgical department for diagnostic and treatment consultation for AFs. The clinical treatment of the cases includes as the first step dietary treatment and hygiene recommendations. The second step is pharmacological treatment trying to achieve chemical sphincterotomy. Finally, the failure of this pharmacological treatment leads to an indication for surgical intervention with a lateral, internal sphincterotomy procedure [3].

Cases: Patients included in the case group satisfied all of the following three criteria: any patient diagnosed with acute or chronic idiopathic AF during the study period, who wished to participate in the study, and who provided a valid written informed consent for this purpose. 

The exclusion criteria for cases were patients with at least one from the following clinical characteristics:AF secondary to other diseases such as tuberculosis, syphilis, herpes, Crohn’s disease, HIV infection, agranulocytosis, carcinoma, leukemia, and hidden abscesses;functional alterations (incontinence or constipation), rectocele, or prolapse;previous surgical treatment for an anorectal disorder or birth trauma; andclinical diagnosis of severe psychiatric disorder, specifically, of depressive disorder or anxiety disorder.

Controls: They were selected from patients who had a surgery consultation for a condition other than anal fissure; and hospital personnel (doctors, nurses, and administrative staff). The controls came from the population from which the cases arose, which minimized possible selection bias [3].

The same exclusion criteria applied to the cases were applied for the controls. Control group participants also provided a valid written informed consent to participate in the study. The controls came from the same population from which the cases arose to minimize possible selection bias.

Comparing cases and controls, the only main difference between both groups was the presence of the inclusion clinical criterion of AFs only in the cases group. Cases are patients diagnosed with acute or chronic idiopathic AFs during the study period.

The final study protocol was approved by the Ethics Committee of San Juan de Alicante University Hospital in 2016 [3].

#### 2.1.3. Variables and Measurements

The instruments used in this study to describe sexuality, QoL, and associated psychological factors in patients with AF were the Spanish, validated versions of the next six questionnaires (Appendix A): Sexuality: CSFQ-14 (Changes in Sexual Functioning Questionnaire, short-form); a higher score relates to better sexual functioning [35,36,37].QoL, health-related: SF-12 (SF-12 Health Survey); a higher score relates to better QoL [38,39,40,41]. MQLI (Multicultural Quality of Life Index); a higher score indicates better QoL [42,43].Associated psychological factors: STAI (State–Trait Anxiety Inventory); a higher score in state and trait anxiety indicates greater anxiety symptoms in state and trait anxiety respectively [44,45,46,47,48]. HADS (Hospital Anxiety and Depression Score); a higher score in anxiety indicates greater anxiety symptoms; a higher score in depression indicates greater depression symptoms [49,50,51]. STAXI-2 (State–Trait Anger Expression Inventory); a higher score in state, trait and expression anger indicates greater anger symptoms in anger state or trait; or a higher level in its expression respectively [52,53,54].

These six questionnaires were all completed by both cases and controls.

All of them have appropriated psychometric characteristics, and the validated Spanish version of each one was used in this study. The internal reliability (Cronbach’s alpha) is provided as one of the main measurement properties of the tools used. Nevertheless, alpha values can differ among studies using different samples.

Some rating scales of each measure are described here by way of example. Each questionnaire may have intrinsic heterogeneity in this regard as well as differences between original and adapted versions. Furthermore, this may vary as it can be associated with some additional extrinsic heterogeneity according to each validation study in specific populations and other related factors.

##### CSFQ-14 (Changes in Sexual Functioning Questionnaire, short-form)

The Changes in Sexual Functioning Questionnaire (CSFQ) [35,36,37], CSFQ-14 is a structured interview/questionnaire designed to measure illness- and medication-related changes in sexual functioning, with evidence of its validity and reliability. The CSFQ is a reliable and valid measure of sexual functioning, useful in both clinical and research settings [35]. The CSFQ-14 is a clinical and research instrument identifying five scales of sexual functioning [36].

The CSFQ-14 is a validated 14-point instrument for measuring sexual function in females and males. The different subdomains are pleasure, desire/frequency, desire/interest, arousal/excitement, and orgasm/completion [35,36]. CSFQ-14 includes CSFQ-P: pleasure, CSFQ-D: desire, CSFQ-E: arousal, CSFQ-O: orgasm, and CSFQ-T: total [35,36]. The CSFQ-14 has been validated in Spanish [37]. Internal reliability (Cronbach’s alpha) was 0.90 [35,36,37].

Scores are reported using a five-point Likert scale that refers to either frequency (“never” to “every day”) or satisfaction (“none” to “great”). For two items (10 and 14), higher sexual functioning corresponds to lower frequency. For all items, higher scores reflect higher sexual functioning. For 12 of the 14 items, higher sexual functioning corresponds to greater frequency or enjoyment/pleasure (e.g., 1 = never to 5 = every day). For two items (item 10, assessing priapism for men and loss of interest after arousal for women; and item 14, assessing painful orgasm), higher sexual functioning corresponds to lower frequency (e.g., 1 = every day to 5 = never). 

Keller et al. [36] describes the five original scales including desire/frequency, a two-item scale assessing frequency of sexual acts, including intercourse and masturbation, and the frequency of desire to participate in sexual activity; desire/interest, a three-item scale that assesses interest in and desire for sexual experiences as expressed in the frequency of sexual thoughts or fantasies and feelings of enjoyment elicited by erotica; arousal/excitement, a three-item scale that assesses frequency of arousal, ease of arousal, and adequate vaginal lubrication/erection during sexual activity; orgasm/completion, a three-item scale that assesses one’s ability to achieve orgasm, including frequency of orgasms, ability to achieve orgasms when desired, and the degree of pleasure derived from orgasm; and pleasure, a single item that assesses current enjoyment of sex life in comparison with past enjoyment [36].

##### SF-12 (SF-12 Health Survey)

The Short-Form 12 Health Survey (SF-12) [38,39,40,41] is a 12-item questionnaire used to assess generic health outcomes from the patient’s perspective. The SF-12 consists of a subset of 12 items from the SF-36 Health Survey (SF-36). The two questionnaires are validated multidimensional instruments measuring the health-related quality of life. Both covering the same eight domains of health outcomes—physical functioning (PF), role-physical (RP), bodily pain (BP), general health (GH), vitality (VT), social functioning (SF), role-emotional (RE), and mental health (MH) [39,40]. The SF-36 and SF-12 questionnaires have both been adapted to Spanish [40,41]. Internal reliability (Cronbach’s alpha) was 0.72 to 0.89. 

It is a tool covering eight domains of health outcomes. It is a self-reported outcome measure used to assess the impact of health on an individual’s everyday life. It is often used as a QoL measure. The SF-12 physical and mental health measures include physical functioning (PF, two items), role limitations due to physical functioning (role-physical (RP), two items), bodily pain (BP, one item), general health (GH, one item) perceptions, vitality (VT, one item), social functioning (SF, one item), role limitations due to emotional problems (role-emotional (RE), two items), and mental health (MH, two items). A higher domain score indicates a better health state.

The SF-12 health survey contains categorical questions (e.g., yes/no) that assess limitations in role functioning as a result of physical and emotional health. This tool also contains Likert response formats including those that are on a three-point scale (e.g., limited a lot, limited a little, or not limited at all) that assess limitations in physical activity and physical role functioning. It also includes a five-point scale (e.g., not at all, a little bit, moderately, quite a bit, and extremely) that assesses pain, and a five-point scale that assesses overall health (excellent, very good, good, fair, and poor). In addition, this tool contains a six-point scale (e.g., all of the time, most of the time, a good bit of the time, some of the time, a little of the time, and none of the time) to assess mental health, vitality, and social functioning.

##### MQLI (Multicultural Quality of Life Index) 

The Multicultural Quality of Life Index [42,43] is a concise instrument for comprehensive, culture-informed, and self-rated assessment of health-related quality of life (QoL). The MQLI has been validated in English, Spanish, and other versions [42,43]. Internal reliability (Cronbach’s alpha) was 0.88 to 0.92 [42,43].

Each of the 10 items is rated on a 10-point Likert scale. A higher score relates to better health-related quality of life (HRQoL). The 10 items or domains are physical well-being (feeling energetic, free of pain and physical problems); psychological/emotional well-being (feeling good, comfortable with yourself); self-care and independent functioning (carrying out daily living tasks, making own decisions); occupational functioning (able to carry out work, school and homemaking duties); interpersonal functioning (able to respond and relate well to family, friends and groups); social emotional support (availability of people you can trust and people who offer help and emotional support); community and service support (good and safe neighborhood, availability of financial resources and other services); personal fulfillment (experiencing a sense of balance, solidarity and empowerment; enjoying sexuality, aesthetics, etc.); spiritual fulfillment (experiencing a high philosophy of life, religiousness, transcendence beyond ordinary material life); and overall quality of life (feeling satisfied and happy with your life in general).

##### STAI (State–Trait Anxiety Inventory) 

The State–Trait Anxiety Inventory (STAI) [44,45,46,47,48] is a 40-item self-reported psychological test for adults, designed to measure state and trait anxiety, with its central distinction between anxiety as an immediate emotional state versus anxiety as a personality trait [47]. The state anxiety are the feelings of immediate anxiety at the current moment. Trait anxiety is dispositional anxiety [44,47]. The STAI is the leading measure of anxiety worldwide [46]. STAI includes A-State, A-Trait with a validated Spanish version [48]. Internal reliability (Cronbach’s alpha) was 0.90 for trait anxiety and 0.94 for state anxiety.

The STAI measures two types of anxiety: state anxiety, as transient current anxiety; and trait anxiety, as a personal characteristic. In the Spanish adaptation, the response scale ranges from 0 to 3 points, by contrast, the original STAI ranges from 1 to 4 points. In both types, higher scores are positively correlated with higher levels of anxiety.

Anxiety state is the perception of transitory threat or displeasure; it refers to feelings of apprehension, tension, nervousness, or worry; it can be described as a temporal–emotional cross-section in the life of a person. Trait anxiety is a personality disposition such as a tendency to perceive situations as threatening and experiencing state anxiety in stressful situations.

Items 1–20 measure state anxiety (STAI-S), and items 21–40 measure trait anxiety (STAI-T) [44,45,46,47,48]. For the state items, respondents are asked to indicate “How you feel right now, that is, at this moment” responses indicate intensity of feeling on a 1 to 4 scale, from “not at all” through “somewhat”, “moderately so”, to “very much so.” For the trait items, the question concerns “how you generally feel” and the response scale indicates frequency: “almost never”, “sometimes”, “often”, and “almost always”.

##### HADS (Hospital Anxiety and Depression Scale) 

The HADS [49,50,51,52] is a self-assessment scale, and it was originally designed to detect depression and anxiety, two common mental disorders, in adults aged 16–65 years attending medical (nonpsychiatric) outpatient clinics.

There are two subscales, the depression subscale (HADS-D) and the anxiety subscale (HADS-A), consisting of seven items each. The HADS is a very well-established scale to detect anxiety and depression in different settings, mostly medical ones. The scale is being used both in research and in clinical practice. The HADS is being using as a screening tool of depression and anxiety and as a measure of their severity. This scale is one of the most highly used and cited symptom scales in psychiatry. HADS has been validated in a Spanish population [52]. Internal reliability (Cronbach’s alpha) for HADS-A ranged from 0.68 to 0.93 (mean 0.83) and for HADS-D from 0.67 to 0.90 (mean 0.82). 

The HADS depression subscale is based on the absence of positive affect. The HADS anxiety subscale is related to worry or cognitive aspects of anxiety. In both subscales, items use a four-point Likert scale ranging from 0 (“not at all”) to 3 (“most of the time”). Reverse scoring is used for items with positive wording. A higher score indicates greater anxiety and greater depression symptoms.

##### STAXI-2 (State–Trait Anger Expression Inventory) 

The STAXI-2 [53,54,55] state anger scale measures the intensity of anger as an emotional state at a particular time. The STAXI-2 trait anger scale assesses how often angry feelings are experienced over time. The STAXI-2 scales and subscales are as follows: six scales, five subscales, and an Anger Expression Index providing an overall measure of total anger expression. State Anger (S-Ang): state anger/feeling (S-Ang/F), state anger/physical (S-Ang/P), state anger/verbal (S-Ang/V); Trait Anger (T-Ang): trait anger/temperament (T-Ang/T), trait anger/reaction (T-Ang/R). AX Index: anger expression—out (AX-O), anger expression—in (AX-I), anger control—out (AC-O), anger control—in (AC-I), EI: index of the frequency with which anger is expressed [55]. 

STAXI-2 measures the experience, expression, and control of anger. Ratings of items are on four-point response scales that measure state anger (intensity) as well as trait anger (frequency). A higher score in every item of each subscale indicates greater anger symptoms intensity (state anger) and the frequency at which anger is experienced (trait anger), expressed (anger expression), and controlled (anger control). STAXI-2 includes six scales, five subscales (three for anger state and two for anger trait), and a final Anger Expression Index providing an overall measure of total anger expression, with a validated Spanish version. Internal reliability (Cronbach’s alpha) was from 0.82 to 0.93 for the complete inventory, from 0.89 to 0.97 for trait anger, and from 0.84 to 0.90 for state anger.

The state anger scale assesses the intensity of anger experienced at a specific time. It includes feeling angry (anger ranging from feeling annoyed to furious); feel like expressing anger verbally (yelling or shouting); and feel like expressing anger physically (hitting someone or breaking things).

The trait anger scale assesses how often angry feelings are experienced over time in the tendency to perceive a wide range of situations as annoying or frustrating and the disposition to respond to such situations with elevations in S-Anger. This includes angry temperament as tendency to experience and express anger indiscriminately and angry reaction as disposition to express anger when criticized or treated unfairly by others. 

The anger expression and anger control scales measures four anger-related traits: anger expression—out (expression of anger toward other persons or objects in the environment); anger expression—in (holding in or suppressing angry feelings); anger control—out (controlling angry feelings by preventing the expression of anger toward other persons or objects in the environment); and anger control—in (controlling suppressed angry feelings by cooling off or calming down). 

STAXI-2 scoring: Higher scores on the trait anger scale are indicative of a higher predisposition to anger, and higher scores on the state anger scale are indicative of higher levels of anger while completing the inventory. Higher scores in angry temperament reflect greater likelihood to become angry, independent of the provocation. Higher scores in angry reaction reflect a tendency to become angry in response to criticism or unfair treatment. Higher scores on the anger—in scale indicate the individual is more likely to suppress anger. Higher scores on the anger—out scale indicate greater likelihood of directing the anger toward a person or object in the environment. Higher scores on the anger control scale reflect more attempts to control the expression of anger. Higher scores on the anger expression scale are indicative of more expressed anger, regardless of whether it is suppressed (anger—in) or directed toward an object (anger—out).

#### 2.1.4. Sample Size Calculation 

A previous pilot study was implemented with 14 cases and 14 controls to estimate the mean score of the questionnaires used.

Sample size calculation was performed in a previous paper to assess the association between neuroticism and anal fissures. In that work, we determined that with an effect size of 1.073 (t-test with an allocation ratio of 1:1), we achieved a power of contrast around 99% [3]. Consequently, this a secondary study of that paper. With these data, it was found that at least 32 cases and 32 controls were needed, assuming a type I error of 5%, a power of 99%, and an allocation ratio of one control per case [3]. 

#### 2.1.5. Statistical Methods—Data Analyses 

The variables were described using absolute and relative frequencies (qualitative) and means with standard deviation (quantitative). The means of the various measurements between cases and controls from the questionnaire were compared using the Student t test. The ORs (odds ratios) were adjusted for age and sex and were obtained for each of the measurements, determining goodness-of-fit (i.e., how well the model fit the data using the Hosmer–Lemeshow test) and the discriminatory capacity of the model (AUC, the area under the receiver operating characteristic curve). The dependent variable was the group (case vs. control) and the independent ones were age, gender, and the outcome. We performed a logistic regression model for each outcome [3].

All the models used were fully consistent with the data (the Hosmer–Lemeshow test did not reach statistical significance in any of the cases). The H–L test indicates the calibration of the logistic regression model (whether the expected and observed probabilities of AFs are similar). This would be when *p* > 0.05. The AUC was used to measure the discriminatory power of the model (ranging: 0.5–1). Values around 0.7 were considered to be satisfactory. If we obtain a good calibration and discrimination, the results will correctly explain the association between AF and the personal psychological characteristics studied.

No missing data were observed in all the collected variables in this work.

All analyses were performed with a significance of 5%. The associated CI (confidence interval) was calculated for each relevant parameter. All calculations were performed in IBM SPSS Statistics 25 and R 3.3.3.

#### 2.1.6. Ethics

This study was performed according to the latest version of the Declaration of Helsinki. The Hospital of San Juan (Alicante, Spain) Clinical Research Ethics Committee approved the study, and signed informed consent was obtained from all participants.

## 3. Results

The average scores obtained by the participants in the instruments applied in the study are shown in Table 1. In the STAI, the averages were 19.1 in state and 16.0 in trait. In the HADS, the averages were 7.0 in anxiety and 4.2 in depression. In the SF-12 questionnaire, the averages were 4.9 in physical functioning, 3.3 in physical role, 2.6 in bodily pain, 3.0 in general health, 3.3 in vitality, 3.8 in social functioning, 3.6 in emotional role, and 7.7 in mental health. In the CSFQ-14 they obtained averages of 43.7 in total sexual functioning, 2.8 in pleasure, 14.5 in desire, 10.9 in arousal, and 10.5 in orgasm. In the STAXI-2 the averages recorded were 17.1 in state anger, 6.2 in state anger/feeling, 5.3 in state anger/physical, 5.8 in state anger/verbal, 17.5 in trait anger, 7.3 in trait anger/temperament, 11.0 in trait anger/reaction, 9.9 in anger expression—out, 12.1 in anger expression—in, 17.8 in anger control—out, 15.0 in anger control—in, and 24.9 in index of the frequency with which anger is expressed. In the MQLI, the average score of the participants in quality of life was 79.8.

The analysis of the differences between cases and controls (Table A1 and Table A2 in Appendix B) revealed an average score significantly higher for the cases in anxiety state, in anxiety trait, in HADS—anxiety, in HADS—depression, in bodily pain, in general health, in vitality, in state anger, in state anger/feeling, in trait anger and in trait anger/reaction; while, the comparison between the groups showed an average score that was significantly higher for the controls, in physical functioning, in role physical, in social functioning, in total sexual functioning, in pleasure, in orgasm, and in quality of life. It was also observed that, although the cases obtained an average score higher in the trait anger/temperament dimension than the controls, this difference was not statistically significant, but it was very near statistical significance. In the same way, the highest average obtained by the controls did not reach statistical significance in emotional role, but it was very near statistical significance. On the other hand, the analysis of the participants’ scores in the rest of the dimensions studied indicated that there were no statistically significant differences between the averages of the group of cases and those of the controls in the traits of mental health, desire, arousal, state anger/physical, state anger/verbal, anger expression—out, anger expression—in, anger control—out, anger control—in, and in the index of the frequency with which anger is expressed. The values of these variables in the total sample, in both cases and controls, are specified in Table A1 in Appendix B.

The setting of the results by sex and age (Table A2 and Table A3 in Appendix B) was significant in the majority of the dimensions analyzed: (1) the group of cases revealed more anxiety state, more anxiety trait, more HADS—anxiety, more HADS—depression, more bodily pain, more general health, more vitality, more state anger/feeling, more trait anger and more trait anger/reaction; and (2) the group of controls showed more physical functioning, more role physical, more social functioning, more sexual functioning total, more pleasure, more desire, more orgasm, and more quality of life. Discrimination values of area under the ROC curve (AUC) (Table A2) of 0.70 or more were obtained in bodily pain, with this being the trait with the highest discrimination value, followed by the dimensions of role physical, physical functioning, anxiety state, pleasure, orgasm, state anger/feeling, quality of life, sexual functioning total, anxiety trait, general health, social functioning, trait anger, desire, vitality, and trait anger/reaction. 

## 4. Discussion

Previous studies have focused on the relationship between medical diseases or mental disorders including its treatments and patients sexuality, QoL (general or health related), or selected psychological factors highlighting the relationship between these factors and clinical disorders or its treatments [5,10,11,13,14,15,19,20,23,25,29,34,56,57,58,59,60,61,62,63,64,65]. These include, sex and quality of life [10]; sexual health and dysfunction in patients with rheumatoid arthritis [13]; sexuality and mental health [15]; antipsychotic-related sexual dysfunction [19]; sexual function in chronic illness [20]; the impact of physical illness on sexual dysfunction [23]; sexual dysfunction and chronic illness [27]; sex and chronic physical illness [28]; comorbidities in male and female sexual dysfunction [60]; sexual dysfunction and mental health in patients with multiple sclerosis and epilepsy [29]; psychiatric disorders and sexual dysfunction [63]; thyroid autoimmune disease impacting on sexual function in young women [64]; clinical features associated with female genital mutilation/cutting [65]; QoL after flatfoot surgery [33]; anxiety and QoL in patients with type 2 diabetes [58]; QoL and associated psychological distress in patients with knee arthroplasty [59]; psychological factors as determinants of medical conditions [57]; sleep disturbance, depression and anxiety in frail patients with atrial fibrillation [61]; cognitive behavior counseling in preoperative preparation and enhanced recovery after surgery [62], between others studies of interest for clinicians.

To our knowledge, there are few studies carried out about sexological aspects, quality of life, or behavioral factors in patients diagnosed with AFs [1,2,3,4,7,8,9,26,66,67]. Selected studies and issues specifically related to anal fissure (AF) and associated factors include prevention and quality of life in anal fissure [1]; anal fissure and personality traits [3]; recurrent anal fissure [2]; stress, psychopathology, and QoL in chronic anal fissure (CAF) patients [4]; QoL in patients with chronic anal fissure (CAF) after topical treatment with diltiazem [5]; fistula-in-ano as anal condition [8]; STDs as anal conditions [7]; sexually transmitted infections of the anus and rectum [9]; QoL and botulinum A toxin treatment for anal fissure [26]; sexual dysfunction in patients with anal fistulas and anal fissures [66]; QoL in patients with chronic anal fissure (CAF) after sphincterotomy [67], and other clinical problems related to anal fissures. 

A frequent drawback of previous studies is the limitation of the factors analyzed in relation to AFs. Studies on behavioral factors associated with anal fissures usually consider only some isolated or scarce behavioral aspects of interest. 

Both the cross-cultural validation of the instruments and the consideration of the cultural and geographical characteristics of the populations and their samples are important for an adequate interpretation of the results of the studies; these could be compared with the results of other samples from different cultures and geographical regions. The sample of this study comes from the same population for both cases and controls. It is interesting to have comparative studies of different countries and cultures, which allow obtaining cultural adaptations that can contribute to a better cross-cultural understanding of the results.

The results of the current study include a broader spectrum of comorbid behavioral conditions significantly associated with the diagnosis of AF. This study could be a significant contribution to the scarce specific research literature with special clinical interest on this topic.

The present study analyzed whether sexual function; QoL; and selected psychological factors such as anxiety, depression, and anger in patients diagnosed with AF compared with the control group were significantly different or not. 

The surgical preoperative preparation of patients [62] including the expanded evaluation embracing sexuality, QoL, and associated psychological factors and the intervention on these can be a fundamental part of a patient’s enhanced recovery [62].

Preoperative anxiety is often associated with poor surgical outcomes. Cognitive behavioral therapy (CBT) can be effective in changing patient behavior toward their own better long-term health. CBT within multimodal interventions is oriented to improve patients’ whole well-being prior to surgery [62].

Preoperative patient education includes psychological support and appropriate information. The key benefits of preoperative education include reduced anxiety, pain, and complications, improving patient compliance, satisfaction, and outcomes [68]. 

Clinical consideration of surgical patients [2] diagnosed with AF [1,2,3,4,7,8,9,26,66,67] or other diagnoses can be better perceived from a person-centered approach model of the patient as a whole person [69] with a global clinical overview, including relevant psychological factors as in the current study. 

The significant factors associated with each group are as follows: 

Higher values in cases than in controls. In STAI: both anxiety state and trait. HADS: both anxiety and depression. SF-12: bodily pain, general health, vitality. STAXI-2: state anger, feeling and verbal. Trait anger, temperament and reaction. Anger expression—out, anger expression—in, anger control—out. IEI: index of the frequency with which anger is expressed.

Higher values in controls than in cases. SF-12 PF: physical functioning, role physical, social functioning, role emotional. CSFQ: total, pleasure, desire, arousal, orgasm, and also, in MQLI.

Globally, the positive and healthy factors were higher in controls than in cases in the following: QoL physical functioning, role physical, social functioning, and role emotional (SF-12); and MQLI. As well all sexuality factors (CSFQ-14): pleasure, desire, arousal, orgasm, and total.

Higher values in cases that in controls. Anxiety both state and trait (STAI). Anxiety and depression (HADS). STAXI-2: anger: state, feeling, and verbal. Trait, temperament and reaction. Anger expression, out and in. Anger control out. IEI (index anger expressed).

Surprisingly, general health (GH) and vitality (V) (SF-12) were higher in cases that in controls.

These results confirm our H1: There are significant differences between case and control groups as mentioned before and these were highlighted by the results as shown in Table A1 and Table A2. If our results are confirmed, we should implement multidisciplinary teams including psychological assessments and interventions on AF patients for increasing therapeutic effectiveness and to better outcomes.

### Strengths, Limitations, and Future Research

This study evaluated a series of clinical aspects in patients with AFs using psychometric instruments validated both in the original samples and in Spanish samples. Our study can contribute to a further understanding of how AFs might influence sexuality and associated relevant psychological factors as sexuality, QoL, anxiety, depression, and anger.

However, some limitations should be highlighted. This cross-sectional study design does not allow to establish causal relationships between its various factors. All data used in this study were collected from individuals who were voluntarily attending treatment and agreed to participate. The generalizability of our results to the entire population of patients with AF is limited. The absence of external validity does not allow the results of this study to be generalized to other samples or populations. 

Additionally, there are some limitations derived from the use of self-report questionnaires. The absence of validated questionnaires used for these objectives with prior standardization based on the criteria from current main taxonomies (DSM and ICD systems) [70,71] should be considered a relevant limitation of this type of study.

Future studies could obtain more specific conclusions by including appropriate large sample sizes. Oncoming research, especially longitudinal studies, is needed to study in clinical settings the possible causal relationships between anal fissures and patients’ sexuality, QoL, anxiety, depression and anger, and other related psychological factors.

## 5. Clinical Implications

There is a long way to go between the research and clinical fields; especially with regard to human sexuality in the clinical settings [72], which is usually underestimated by clinicians [19] or even ignored by them in their daily clinical practice. QoL [26] and sexual function [66] are rarely studied in patients with surgical diseases, including AFs. Sexuality and mental health should take into consideration by clinicians as part of the patients’ clinical global evaluation and clinical intervention [56]. Addressing the surgical patients’ issues of sexuality, QoL, anxiety, depression, anger, and other associated psychological factors in clinical practice would be beneficial for the clinical assessment and intervention in benefit for every patient considered as a whole person.

## Figures and Tables

**Table 1 jcm-10-04401-t001:** Gender and age variables in Total Sample, Cases and Control groups.

Variables	Total n = 67 n (%)/x ± s	Cases n = 35 (52.2%) n (%)/x ± s	Controls n = 32 (47.8%) n (%)/x ± s
Gender:			
Male	36 (53.7)	22 (62.9)	14 (43.8)
Female	31 (46.3)	13 (37.1)	18 (56.3)
Age (years)	52.2 ± 12.1	52.9 ± 12.7	51.4 ± 11.5

x ± s, mean ± standard deviation.

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
