# Peer review of "Sexuality, Quality of Life, Anxiety, Depression, and Anger in Patients with Anal Fissure. A Case–Control Study"

_jcm, 2021, doi:10.3390/jcm10194401_

Round 1

Reviewer 1 Report

See word document for comments

Author Response

Jcm-1271632 2nd Round

Salamanca (Spain), August 4, 2021.

Dear Editor

Thank you very much for submitting the reviewers' feedback for this jcm- 1271632 manuscript

which will certainly improve the scientific quality of the research work.

Attached you will find the pending responses to the comments of the reviewers # 1 and #3.

Reviewer # 1

The manuscript describes a case-control study which compared patients with anal fissures (AF) to controls on self-report measures of sexual functioning, quality of life, and mental health. Although the authors alluded to the novelty of the current study in the discussion, it was unclear from the introduction how this study added to the scientific literature in this area. Given the literature cited in the introduction on the connection between sexual functioning and quality of life among various patient populations, the readers would benefit from a bit more context upfront and weaved throughout as to why a study was needed in this patient population and setting – in other words, why would you expect different results from the existing studies, and if they don’t, why does this research need to be done? The main conclusion that sexuality and sexual function and mental health should be considered in surgical patients based on the significant findings between the cases and controls, but there seem to be missed opportunities to expand on the discussion. For example, what recommendations do the authors have for addressing the systemic and interpersonal barriers to implement these types of questionnaires considering their comments in the Clinical Implications section and what are the next steps with respect to research and clinical practice?

Specific Comments:

Abstract

Comment 1. Remove the abbreviations from the abstract. It is okay to be more general, such as “validated instruments were used to assess health-related quality of life, anxiety, anger, and sexual function.”

Answer. All the acronyms and abbreviations have been removed. A new sentence has been added as follows (line 37): These instruments were used to assess health-related quality of life, anxiety, anger, depression, and sexual function.

Introduction

Comment 2. Reorganize the introduction so each paragraph flows logically into the next. The topics seem to be scattered and present random information related to the topic.

Answer. Following your suggestion,the introduction has been reorganized and some repetitive sentences about sexuality has been deleted as follows:

Sexual dysfunction (SD) is a common adverse effect associated to treatment with psychotropic drugs; especially antipsychotics and antidepressants.

Patients are sexual beings. Sexuality is a vital dimension of human experience.

Sexual dysfunction as a consequence of treatment with psychotropic drugs is a common adverse effect.

Some new sentences have been added as follows:

Line 71: (HRQoL) measured by a Short Form (SF) Health Survey; including bodily pain, mental health, vitality and general health [6].

Line 125: Anxiety includes two concepts: state anxiety, as immediate and current, and anxiety trait as dispositional. Anger expression is including both anger state and anger trait.

Line 128: Papers covering these aspects are very scarce. This study aims to add to the literature the expected different results profile between case and controls, of clinical interest.

Methods

Comment 3. Remove the table in section 2.2 and refer to the information in Table 2.

Answer. Table of section 2.2 was removed. A new sentence referred to current Table 1 (former Table 2) has been added as follows ( Line 168):Variables of gender (male, female) and age (years) are referred to the total sample, cases and controls (Table 1).

Comment 4. In the description of the measures, please, indicate the direction of the score as it pertains to the construct. For example, a higher score on the SF-12 relates to better quality of life and a higher score on the STAI indicates greater anxiety symptoms.

Answer. A list of scales and subscales indicating the direction of the score is provided in the Appendix A (former Table 1).

Some new information about the scales has been added as follows (lines 210-220):

CSFQ-14 (Changes in Sexual Functioning Questionnaire short-form).A higher score relates to better sexual functioning.

SF-12 (SF-12 Health Survey) A higher score relates to better QoL.

MQLI (Multicultural Quality of Life Index). A higher score indicates better QoL

STAI (State-Trait Anxiety inventory). A higher score in State and Trait Anxiety indicates greater anxiety symptoms in state and trait anxiety respectively.

HADS (Hospital Anxiety and Depression Scale). A higher score in Anxiety indicates greater anxiety symptoms. A higher score in Depression indicates greater depression symptoms.

STAXI-2 (State-Trait Anger Expression Inventory). A higher score in State, Trait and Expression Anger indicates greater anger symptoms in Anger State or Trait; or higher level in its Expression respectively.

In addition, please, indicate the rating scales of each measure (e.g., 5-point Likert scale, from 1=strongly disagree to 5=strongly agree).

Answer. In order to clarify the information about the scales, several new paragraphs with extended information have been added as follows:

Line 224: Some rating scales of each measure are described here by way of example. Each questionnaire may have intrinsic heterogeneity in this regard as well as differences between original and adapted versions; Furthermore, this may vary as it can be associated with some additional extrinsic heterogeneity according to each validation study in specific populations and other related factors.

Line 230: CSFQ-14 (Changes in Sexual Functioning Questionnaire short-form) [36,37,38]. The tool uses 5-point Likert scales to provide the person an opportunity to self-evaluate sexual behaviors or problems in several areas. It includes CSFQ-T: Total. CSFQ-P: Pleasure. CSFQ-D: Desire. CSFQ-E: Arousal. CSFQ-O: Orgasm.

Scores are reported using a 5-point Likert scale that refers to either frequency (“never” to “every day”) or satisfaction (“none” to “great”). For two ítems (10 and 14), higher sexual functioning corresponds to lower frequency. For all items, higher scores reflect higher sexual functioning. For 12 of the 14 items, higher sexual functioning corresponds to greater frequency or enjoyment/pleasure (e.g., 1 = never to 5 = every day). For two ítems (item 10, assessing priapism for men and loss of interest after arousal for women; and item 14, assessing painful orgasm), higher sexual functioning corresponds to lower frequency (e.g., 1 = every day to 5 = never).

Keller [37] describes the five original scales including: Desire /Frequency, a 2-item scale assessing frequency of sexual acts, including intercourse and masturbation, and the frequency of desire to participate in sexual activity; Desire /Interest, a 3-item scale that assesses interest in and desire for sexual experiences as expressed in the frequency of sexual thoughts or fantasies and feelings of enjoyment elicited by erotica; Arousal /Excitement, a 3-item scale that assesses frequency of arousal, ease of arousal, and adequate vaginal lubrication /erection during sexual activity; Orgasm /Completion, a 3-items cale that assesses one’s ability to achieve orgasm, including frequency of orgasms, ability to achieve orgasms when desired, and the degree of pleasure derived from orgasm; and Pleasure, a single item that assesses current enjoyment of sex life in comparison with past enjoyment [37].

Line 300: SF-12 (SF-12 Health Survey): It is a tool covering eight domains of health outcomes. It is a self-reported outcome measure used to assess the impact of health on an individual's everyday life. It is often used as a QoL measure. The SF-12 physical and mental health measures includes: physical functioning (PF, two items),role limitations due to physical functioning (role-physical [RP],two items), bodily pain (BP, one item), general health (GH, one item) perceptions, vitality (VT, one item), social functioning (SF,one item), role limitations due to emotional problems (role-emotional [RE], two items), and mental health (MH, two items). A higher domain score indicates a better health state.

The SF-12 health survey contains categorical questions (e.g., yes/no) that assess limitations in role functioning as a result of physical and emotional health. This tool also contains Likert response formats including those that are on a three-point scale (e.g., limited a lot, limited a little, or not limited at all) that assess limitations in physical activity and physical role functioning. Also are included a five-point scale (e.g., not at all, a little bit, moderately, quite a bit, and extremely) that assesses pain, and a five-point scale that assesses overall health (excellent, very good, good, fair, and poor). In addition this tool contains a six-point scale (e.g., all of the time, most of the time, a good bit of the time, some of the time, a little of the time, and none of the time) to assess mental health, vitality, and social functioning.

Line 326: MQLI (Multicultural Quality of Life Index). It is composed of 10 items (from physical well-being to global perception of quality of life). Each item is rated on a 10-point Likert scale. A higher score relates to better Health Related Quality of Life (HRQoL). The 10 items or domains are: Physical well-being (feeling energetic, free of pain and physical problems); Psychological/emotional well-being (feeling good, comfortable with yourself); Self-care and independent functioning (carrying out daily living tasks, making own decisions);  Occupational functioning (able to carry out work, school and homemaking

duties); Interpersonal functioning (able to respond and relate well to family, friends and groups); Social emotional support (availability of people you can trust and people who offer help and emotional support); Community and service support (good and safe neighborhood, availability of financial resources and other services); Personal fulfillment (experiencing a sense of balance, solidarity and empowerment; enjoying sexuality, aesthetics, etc.); Spiritual fulfillment (experiencing a high philosophy of life, religiousness, transcendence beyond ordinary material life); and Overall quality of life (feeling satisfied and happy with your life in general)

Line 353: STAI (State-Trait Anxiety inventory). The STAI measures two types of anxiety; state anxiety, as transient current anxiety; and trait anxiety, as a personal characteristic. In the Spanish adaptation, the response scale ranges from 0 to 3 points, by contrast with the original STAI ranging from 1 to 4 points. In the two types higher scores are positively correlated with higher levels of anxiety.

State anxiety reflects how threatening a person perceives his environment being transitory unpleasant; it refers to feelings of apprehension, tension, nervousness or worry; it can be described as a temporal emotional cross-section in the life of a person. Trait anxiety is a personality disposition as tendency to perceive situations as threatening; and experiencing state anxiety in stressful situations.

Items 1-20 measure state anxiety (STAI-S), and items 21-40 measure

trait anxiety (STAI-T). For the state items, respondents are asked to indicate “How you feel right now, that is, at this moment.” Responses indicate intensity of feeling on a 1 to 4 scale, from “not at all” through “somewhat”, moderately so” to “very much so.” For the trait items the question concerns “how you generally feel” and the response scale indicates frequency: “almost never”, “sometimes”, “often” and “almost always.”

Line 377: HADS (Hospital Anxiety and Depression Scale).

There are two subscales. The Depression subscale, based on the absence of positive affect; and the Anxiety subscale, related to worry or cognitive aspects of anxiety. In both subscales items uses a 4-point Likert scale ranging from 0 (as “not at all”) to 3 (as “most of the time”). Reverse scoring is used for items with positive wording. A higher score indicates greater Anxiety and greater Depression symptoms severally.

Line 394: STAXI-2 (State-Trait Anger Expression Inventory). It measures the experience, expression, and control of anger. Ratings of items are on a 4-point response scales that measure state anger (intensity) as well as trait anger (frequency). A higher score in every item of each subscale indicates greater anger symptoms intensity (State Anger); and the frequency that anger is experienced (Trait Anger), expressed (Anger Expression), and controlled (Anger control). STAXI-2 includes six scales, five subscales (three for Anger State and two for Anger Trait), and an final Anger Expression Index providing an overall measure of total anger expression.

The State Anger scale assesses the intensity of anger experienced at a specific time. It includes: Feeling Angry (Anger ranging from feeling annoyed to furious);

Feel like Expressing Anger Verbally (yelling or shouting); and Feel like Expressing Anger Physically (hitting someone or breaking things).

The Trait Anger scale assesses how often angry feelings are experienced over time in the tendency to perceive a wide range of situations as annoying or frustrating and the disposition to respond to such situations with elevations in S-Anger. Includes Angry Temperament as tendency to experience and express anger indiscriminately; and Angry Reaction as disposition to express anger when criticized or treated unfairly by others.

The Anger Expression and Anger Control scales measures four anger-related traits: Anger Expression-Out (expression of anger toward other persons or objects in the environment); Anger Expression-In (holding in or suppressing angry feelings); Anger Control-Out (controlling angry feelings by preventing the expression of anger toward other persons or objects in the environment); and Anger Control-In (controlling suppressed angry feelings by cooling off or calming down).

Scoring. Higher scores on the Trait-Anger scale are indicative of a higher predisposition to anger, and higher scores on the State Anger scale are indicative of higher levels of anger while completing the inventory. Higher scores on Angry Temperament reflect greater likelihood to become angry, independent of the provocation. Higher scores on Angry Reaction reflect a tendency to become angry in response to criticism or unfair treatment. Higher scores on the Anger-In scale indicate the individual is more likely to suppress anger. Higher scores on the Anger-Out scale indicate greater likelihood of directing the anger towards a person or object in the environment. Higher scores on Anger Control scale reflect more attempts to control the expression of anger. Higher scores on the Anger Expression scale are indicative of more expressed anger, regardless of whether it is suppressed (anger-in) or directed toward an object (anger-out).

Comment 5. Within the authors’ description of each questionnaire, the readers would benefit from an operational definition for each construct and subscale. This would help interpretation of the results as well. For example, some of the STAXI-2 subscales were significantly different between the cases and controls while others were not. Without the authors describing what “state anger (S-Ang)” or “State Anger/Physical” means, we don’t really understand what it means when there is a difference in one but not the other.

Answer. The new information is included in the response to previous comment 4.

Comment 6. It was unclear whether the AF case group were being seen at the hospital for pre-operative consultation for AF, being treated medically for AF, or some combination thereof. Please provide more information about the clinical treatment status of the cases.

Answer. In orther to clarify this point a new paragraph has been added as follows: (line 175)

Patients of the case group were being seen at the hospital surgical department for diagnostic and treatment consultation for AF. The clinical treatment of the cases includes as the first step dietary treatment and hygiene recommendations. The second step is the pharmacological treatment trying to achieve chemical sphincterotomy. Finally the Failure of this pharmacological treatment leads to an indication for surgical intervention with a lateral internal sphincterotomy procedure [3].

Comment 7. Were controls matched to controls regarding age and sex? Please describe in more detail how they were selected.

Answer. The controls were not matched by gender or age, consequently, we have adjusted our results by these confounders. In fact, you can see that there are relevant differences in gender in both groups.

A new sentence has been added to clarify this point as follows (line194): The controls came from the population from which the cases arose, which minimized possible selection bias.

Comment 8. Provide the effect sizes used to calculate the power analyses (section 2.4 Sample Size Calculation).

Answer. The sample size calculation was calculated for a previous study which we have referenced [3]. Therefore, we have edited the text in accordance adding a new paragraph as follows (line 446): Sample size calculation was performed in a previous paper to assess the association between neuroticism and anal fissure. In that work, we determined that with an effect size of 1.073 (t-test with an allocation ratio of 1:1), we achieved a power of the contrast around 99% [3]. Consequently, this a secondary study of that paper.

A final part of the sentence “… a power of 99%, and an allocation ratio of one control per case” has been deleted

Comment 9. Were there missing data and how were they handled?

Answer. We did not have any missing data. A new sentence to clarify that has been added as follows (line 469): “No missing data was observed in all the collected variables in this work.”

Results

Comment 10. Why did the authors choose to do multivariable logistic regressions to control for age and sex, particularly when they did not find differences in these two factors by cases and controls (the outcome in the models)?

Answer. Age and gender could be relevant factors when we are talking about the outcomes which we have assessed, therefore we think that an adjustment is needed. Furthermore, although we did not find any differences between the groups, we could appreciate a relevant difference in gender. Finally, to be sure that all these points had not influenced the results, we decided to perform this multivariate analysis.

Comment 11. It would be more meaningful to know if there were differences in sex and/or age in the constructs themselves (correlation between age and each scale, and t-tests between sex and each scale). If differences/significant associations are found, the authors should do ANCOVAs instead of the logistic regressions with sex, age, and group (case & control) as independent variables and the questionnaire scales as the outcomes.

Answer. Thanks for your interesting approach, nevertheless our study has a case-control design in which is necessary to assess association measures and, more particularly, odds ratios. Nonetheless, we have wanted to provide that information in a descriptive way, because we consider that it could be relevant for readers (Table 1). That kind of analysis could be relevant to be performed in a cross-sectional study, where the prevalence of anal fissure is similar with the general population. This information has been included as a new future line research in the discussion.

Comment 12. The text of the results is far too repetitive of what is already presented in the tables. Please reduce the repetitive information and refer to the table(s) for exact numbers.

Answer. Following your suggestion, the text of the results has been reduced and a new sentence has been added as follows: The values of these variables in the total sample, in cases and in controls are specified on the table 1.

*Please note that the Current Table 1 is the former Table 2

Comment 13. The last paragraph of the results belongs in the Methods under a subheading such as “Data Analyses.”

Answer: This paragraph has been moved to 2.5. Statistical Methods. Data Analyses.

Comment 14. Discussion

The summary repeating the findings should be cut down and the discussion should be bulked up significantly to discuss how these results relate to the existing literature and go into more detail on the meaning of the findings and their implications. The discussion in its current state is far too shallow.

Answer. In order to clarify and increase the information, some new references have been added in the discussion as follows:

Line 553: This include, Sex and Quality of Life [11]; Sexual Health and Dysfunction in Patients With Rheumatoid Arthritis [14]; Sexuality and Mental Health [16]; Antipsychotic-Related Sexual Dysfunction [20]; Sexual function in chronic illness [21]; The impact of physical illness on sexual dysfunction [24]; Sexual dysfunction and chronic illness [28]; Sex and Chronic Physical Illness [29]; Comorbidities in Male and Female Sexual Dysfunction [61]; Sexual dysfunction and mental health in patients with multiple sclerosis and epilepsy [30]; Psychiatric disorders and sexual dysfunction [64]; Thyroid Autoimmune Disease Impacting on Sexual Function in Young Women [65]; Clinical Features Associated with Female Genital Mutilation /Cutting [66]; QoL after Flatfoot Surgery [34]; Anxiety and QoL in Patients with Type 2 Diabetes [59]; QoL and Associated Psychological Distress in Patients with Knee Arthroplasty [60]; Psychological Factors as Determinants of Medical Conditions [58]; Sleep Disturbance, Depression and Anxiety in Frail Patients with atrial fibrilation [62]; Cognitive Behavior Counseling in Preoperative Preparation and Enhanced Recovery After Surgery [63], between others studies of interest for clinicians.

To our knowledge, there are few studies carried out about sexological aspects, quality of life or behavioral factors in patients diagnosed of AF [1,3,4,5,8,9,10,27,67,68]. Selected studies and issues specifically related to Anal Fissure (AF) and associated factors include: Prevention and Quality of Life in Anal Fissure [1]; Anal Fissure and Personality traits [3]; Recurrent Anal Fissure [4]; Stress, psychopathology and QoL in chronic anal fissure (CAF) patients [5]; QoL in patients with chronic anal fissure (CAF) after topical treatment with diltiazem [6]; Fistula-in-Ano as Anal Condition [9]; STDs as Anal Conditions [8]; Sexually transmitted infections of the anus and rectum [10]; QoL and Botulinum A Toxin Treatment for Anal Fissure [27]; Sexual dysfunction in patients with anal fistulas and anal fissures [67]; QoL in patients with chronic anal fissure (CAF) after sphincterotomy [68], and other clinical problems related to Anal Fissure.

A frequent drawback of previous studies is the limitation of the factors analyzed in relation to AF. Studies on behavioral factors associated with anal fissure usually consider only some isolated or scarce behavioral aspects of interest.

The results of the present study include a broader spectrum of comorbid behavioral conditions significantly associated with the diagnosis of AF. This study could be a significant contribution to the scarce specific research literature with special clinical interest on this topic.

Comment 15. How might the geographical and/or cultural setting affect the interpretation of the findings?

Answer. A new paragraph has been added as follows (line 581):

Both the cross-cultural validation of the instruments and the consideration of the cultural and geographical characteristics of the populations and their samples are important for an adequate interpretation of the results of the studies; these could be compared with the results of other samples from different cultures and geographical regions. The sample of this study comes from the same population in both cases and controls. It is interesting to have comparative studies of different countries and cultures, which allow obtaining cultural adaptations that can contribute to a better cross-cultural understanding of the results.

Comment 16. Table 1 and 4 are unnecessary. You may use an appendix to list the abbreviations. There may be a better way to communicate the information in Table 4, such as a figure of the means by cases and controls

Answer. The abbreviations are now listed in Appendix 1, (former Table 1). Table 4 has been deleted and its contents are being explained in the Discussion as follows:

Higher values in cases than in controls. In STAI: Both Anxiety State and Trait. HADS: Both Anxiety and Depression. SF-12: Bodily Pain, General Health, Vitality. STAXI-2: State Anger, / Feeling, / Verbal. Trait Anger, / Temperament / Reaction. Anger Expression – Out, Anger Expression – In. Anger Control – Out. IEI: Index of the frequency with which anger is expressed.

Higher values in controls than in cases. In: SF-12 PF: Physical Functioning, Role Physical, Social Functioning, Role Emotional. CSFQ: Total, Pleasure, Desire, Arousal, Orgasm and also, in MQLI.

Globally, the positive and healthy factors were higher in controls than in cases in: QoL Physical Functioning, Role Physical, Social Functioning and Role Emotional (SF-12); and MQLI. As well all sexuality factors (CSFQ-14): Pleasure, Desire, Arousal, Orgasm and Total.

Higher values in cases that in controls. In: Anxiety both state and trait (STAI). Anxiety and Depression (HADS). In (STAXI-2): Anger: State, Feeling, Verbal. Trait, Temperament, Reaction. Anger Expression Out and In. Anger Control Out. IEI (Index anger expressed).

.................

Reviewer 2 Report

Authors responded clearly to the critical issues identified.

Author Response

No additional comments

Reviewer 3 Report

*Introduction

-Ideas in the introduction seem to require more work. For instance, there is a lot devoted to sex (even some repetitive ideas) while almost nothing to anger

-It is not clear what is the contribution of this study.

*Materials and methods.

-Some inaccurate terms (e.g., 163: in this design you cannot "assign to" a group. This is a non-manipulative design.

-Date of study repeated (lines 150 and 165).

-Although cases and controls are applied the same exclusion criteria, it is not clearly justified their comparability (other than age and sex)

-Sample size: It is not clear in this manuscript nor in the original study (Luri-Prieto, et al) how the sample size was calculated (what was Expected OR: between exposed and non-exposed groups, the probability of exposure in cases or the probability of exposure in control), or for what test or the tool used to calculate it. 

-No evidence regarding measurement propoerties of the tools used is provided (e.g., internal consistency).

-Data analysis lack some information: How the adjustied OR were obtained? It was logistic regression for every response variable? What terms were entered?

*Results

-Readibility is impaired by repeating in the text what is already in table 2 and 3.

-How was Table 4 obtained?

Author Response

Jcm-1271632 2nd Round

Salamanca (Spain), August 5, 2021.

Dear Editor

Thank you very much for submitting the reviewers' feedback for this jcm- 1271632 manuscript

which will certainly improve the scientific quality of the research work.

Attached you will find the pending responses to the comments of the reviewer #3.

Responses to Reviewer # 3

Comment 1. Introduction. Ideas in the introduction seem to require more work. For instance, there is a lot devoted to sex (even some repetitive ideas) while almost nothing to anger

Answer,

Reviewing the introduction some repetitive sentences about sexuality are deleted as: Sexual dysfunction (SD) is a common adverse effect associated to treatment with psychotropic drugs; especially antipsychotics and antidepressants.

Patients are sexual beings.

Sexuality is a vital dimension of human experience.

Sexual dysfunction as a consequence of treatment with psychotropic drugs is a common adverse effect.

To clarify the anxiety concept, a new sentence has been added ( L125)

“Anxiety includes two concepts: state anxiety, as immediate and current, and anxiety trait, as dispositional. Anger expression is including both anger state and anger trait”

Comment 2. It is not clear what is the contribution of this study.

In order to clarify its contribution, a new paragraph has been added as follows (Line 128)

Papers covering these aspects are very scarce. This study aims to add to the literature the expected different results profile between case and controls, of clinical interest.

*Materials and methods.

Comment 3. Some inaccurate terms (e.g., 163: in this design you cannot "assign to" a group. This is a non-manipulative design.

To clarify the terms this sentence has been deleted: The sample consisted of 67 people, 35 of them (52.2%) were assigned to the group of cases and the other 32 to the control group (47.8%).

A new paragraph has been added as follows: (Line 171),

The sample consisted of 67 people, 35 of them (52.2%) were from the group of cases and the other 32 from the control group (47.8%).

Comment 4. Date of study repeated (lines 150 and 165).

This sentence to avoid repeated information has been deleted:The study was performed between January 2016 and February 2017 [3].

Comment 5. Although cases and controls are applied the same exclusion criteria, it is not clearly justified their comparability (other than age and sex)

This sentence has ben deleted: The differential variable between the cases and the controls groups was the presence of anal fissure (AF) diagnostic only in the cases group.

A new paragraph has been added ( Line 201):

Comparing cases and controls, the only main difference between both groups was the presence of the inclusion clinical criterion of anal fissure (AF) only in the cases group: Cases are patients diagnosed with acute or chronic idiopathic AF during the study period.

Comment 6. Sample size: It is not clear in this manuscript nor in the original study (Luri-Prieto, et al) how the sample size was calculated (what was Expected OR: between exposed and non-exposed groups, the probability of exposure in cases or the probability of exposure in control), or for what test or the tool used to calculate it. 

Answer: We have only provided more details about the sample size calculation, following your suggestions and those from Reviewer #1. We were interested in determining whether there were differences in means in an outcome (neuroticism) between the two groups (case and control). Furthermore, we have indicated that this is a secondary study. If the reviewer is interested in which we calculate a new sample size (a posteriori of course), we could perform it as well.

Comment 7. No evidence regarding measurement properties of the tools used is provided (e.g., internal consistency).

In order to provide evidence about these properties a new paragraph has been added in chapter 2.3. Variables and Measurements as follows:

The internal reliability (Cronbach's alpha) is provided as one of the main measurement properties of the tools used. Nevertheless, alpha values can differ among studies using different samples

CSFQ-14 (Changes in Sexual Functioning Questionnaire short-form):

Internal reliability (Cronbach's alpha) = 0.90

Health related: SF-12 (SF-12 Health Survey)

Internal reliability (Cronbach's alpha) of 0.72 to 0.89)

MQLI (Multicultural Quality of Life Index)

Internal reliability (Cronbach's alpha) of 0.88 to 0.92)

STAI (State-Trait Anxiety inventory)

Internal reliability (Cronbach’s alpha) of 0.90 for Trait Anxiety and 0.94 for State Anxiety.

HADS (Hospital Anxiety and Depression Scale)

Internal reliability (Cronbach’s alpha) for HADS-A from 0.68 to 0.93 (mean 0.83) and for HADS-D from 0.67 to 0.90 (mean 0.82).

STAXI-2 (State-Trait Anger Expression Inventory)

Internal reliability (Cronbach’s alpha) from 0.82 to 0.93 for the complete inventory; from 0.89 to 0.97 for Trait Anger and from 0.84 to 0.90 for State Anger.

Comment 8. Data analysis lack some information: How the adjustied OR were obtained? It was logistic regression for every response variable? What terms were entered?

We have edited the statistical methods to answer your questions adding a new sentence as follows: (line 452)

“The OR (Odds ratios) were adjusted for age and sex using logistic regression models and were obtained for each of the measurements, determining goodness-of-fit (it is, how well the model fit the data using the test Hosmer-Lemeshow) and the discriminatory capacity of the model (AUC, the area under the receiver operating characteristic curve). The dependent variable was the group (case vs control) and the independent ones were: age, gender and the outcome. We performed a logistic regression model per each outcome.”

*Results

Comment 9. Readibility is impaired by repeating in the text what is already in table 2 and 3.

Answer. In general, redundant information with former tables 2 and 3 (current tables 1 and 2) has been removed from the main text or they have been simplified therein. The long first paragraph has been almost all deleted. The p values referred in the former table 2 are deleted from the text. The p values and OR values referred to the former table 3 are also removed from the text. Finally, the AUC values of the former table 3 are eliminated from the text as well.

A new sentence was added twice as follows: but it was very near to the statistical significance (line 512).

The sentence “Discrimination AUC values (Table 3) of 0.70 or more were obtained in.” has been substituted by this onew one: “Discrimination values of area under the ROC curve (AUC) (Table 2) of 0.70 or more were obtained in…”

Comment 10 How was Table 4 obtained?

Answer:  Finally, Table 4 has been deleted and their contents are explained in the Discussion section.The meaning was indicating the factors which were associated with controls or cases. In other words, if higher scores of a questionnaire were associated with cases, this score appeared in the cases column. This information has been included in its first mention.

.................
